# Recognizing and Looking at Masked Emotional Faces in Alexithymia

**DOI:** 10.3390/bs14040343

**Published:** 2024-04-18

**Authors:** Marla Fuchs, Anette Kersting, Thomas Suslow, Charlott Maria Bodenschatz

**Affiliations:** Department of Psychosomatic Medicine and Psychotherapy, University of Leipzig Medical Center, 04103 Leipzig, Germany; marla.fuchs@gmx.de (M.F.); anette.kersting@medizin.uni-leipzig.de (A.K.); charlott.bodenschatz@medizin.uni-leipzig.de (C.M.B.)

**Keywords:** eye-tracking, visual attention, emotion face, face masks, emotion recognition, alexithymia, sex, gaze behavior

## Abstract

Alexithymia is a clinically relevant personality construct characterized by difficulties identifying and communicating one’s emotions and externally oriented thinking. Alexithymia has been found to be related to poor emotion decoding and diminished attention to the eyes. The present eye tracking study investigated whether high levels of alexithymia are related to impairments in recognizing emotions in masked faces and reduced attentional preference for the eyes. An emotion recognition task with happy, fearful, disgusted, and neutral faces with face masks was administered to high-alexithymic and non-alexithymic individuals. Hit rates, latencies of correct responses, and fixation duration on eyes and face mask were analyzed as a function of group and sex. Alexithymia had no effects on accuracy and speed of emotion recognition. However, alexithymic men showed less attentional preference for the eyes relative to the mask than non-alexithymic men, which was due to their increased attention to face masks. No fixation duration differences were observed between alexithymic and non-alexithymic women. Our data indicate that high levels of alexithymia might not have adverse effects on the efficiency of emotion recognition from faces wearing masks. Future research on gaze behavior during facial emotion recognition in high alexithymia should consider sex as a moderating variable.

## 1. Introduction

An important ability for successful interpersonal interaction in humans is the identification of emotions from facial expressions [1,2,3]. Emotion recognition in faces can rely on attention allocation to facial features, which have high diagnostic relevance for a specific emotion, e.g., the mouth for happiness or the eyes for sadness [4,5]. In addition, holistic processes in face perception seem to be involved in emotion recognition depending on the specific emotion expressed [6,7]. Accuracy of facial emotion recognition is rather high for most of the basic emotions and related to their frequency of occurrence in everyday life [8,9]. Findings from facial emotion recognition tasks show consistently that happiness is the best recognized and fear and sadness are the worst recognized basic emotions [9,10,11]. There is evidence for a female advantage in decoding facial emotions [12,13].

Human social environments underwent a change with the worldwide onset of the COVID-19 pandemic. One of the measures to prevent the spreading of the coronavirus was the use of face masks [14]. Basic forms of nonverbal communication like facial emotion recognition have been altered by the wearing of face masks. They occlude the lower part of the face, including the mouth and nose, thereby hiding facial features on which humans rely to recognize other people’s emotions [4,15]. Unsurprisingly, recent research revealed that wearing face masks hampers the identification of many basic emotions in facial expressions, e.g., disgust, sadness, happiness, anger, and fear [16,17,18]. Moreover, face masks seem to lead to lower confidence in one’s assessment of the facial emotions displayed by others [19]. The role of individual differences in emotion recognition from occluded faces represents an important research question. The hypothesis that individuals with autistic traits could manifest specific impairments in emotion recognition from occluded faces was not confirmed [20,21]. In the study of Pazhoohi et al. [21], high autistic trait individuals manifested worse emotion recognition than low trait autistic individuals in faces wearing masks but the extent of impairment was similar in the unmasked face condition. Emotional intelligence was not found to be linked to the ability to read emotions from faces with masks [22]. Autistic and alexithymic traits could modulate distinct aspects of face perception: autistic traits were found to be associated with structural encoding of faces whereas alexithymic traits were related to emotion decoding processes [23].

Alexithymia is a personality trait that plays a significant role in the decoding of emotional facial expressions. Alexithymia is defined by difficulties in identifying and verbalizing one’s feelings and tendencies to focus on external events rather than inner experiences [24]. It is thought to be a vulnerability factor for the development and maintenance of mental disorders [25,26]. The prevalence of clinically relevant alexithymia in the general population is approximately 10% [27,28] with a somewhat higher frequency in males [29,30]. In healthy individuals, alexithymia has been found to be associated with impairments in identifying others’ emotional facial expressions when stimuli were presented for a long duration and had high intensity [31,32]. However, some other studies with a long duration of face presentation did not reveal a link between alexithymia and facial emotion recognition [33,34]. Such discrepancies in results could be due to differences in task instructions (the emphasis on accuracy or speed or both). Findings concerning emotion identification impairments in alexithymia are considerably more consistent across studies, which used conditions of suboptimal stimulus presentation, e.g., showing faces in degraded quality or with temporal constraints [35,36,37,38]. It can be anticipated that for high alexithymic individuals, face masks might create an even greater challenge compared to non-alexithymic individuals when facial emotion recognition is required. The systematic review of Grynberg et al. [39] concludes that alexithymic individuals’ impairments in recognizing emotions from facial expressions seem to be neither limited to specific emotional qualities nor a specific emotional valence. This means that alexithymia has been found to be linked to impairments in identifying negative (e.g., angry, disgusted, or fearful) expressions as well as to impairments in identifying positive (e.g., happy) expressions. However, there is evidence from more recent research suggesting that alexithymia could be characterized by pronounced impairments in perceiving and processing fearful [40,41] and other threat-related facial expressions [42]. Disgust is a hostile emotion associated with aggression and conflict [43] that expresses disapproval for the actions of other people [44], whereas facial fear is an indicator of potential indirect threat (danger in the environment) and the expresser’s loss of control [45]. Against this background, it could be of particular interest to study the perception of facial fear and disgust along with that of positive and neutral expressions in alexithymic individuals.

To our knowledge, there are two previous studies that have investigated the relationship between alexithymia and emotion identification in faces wearing face masks. Verroca et al. [46] focused their analysis on recognition accuracy and examined alexithymia as a dimensional construct (using total alexithymia scores). The authors observed no association between overall alexithymia and accuracy in recognizing emotions from masked faces. Maiorana et al. [47] used the facet approach in their analysis but did not find a correlation between alexithymia dimensions and speed of emotion identification in masked faces. In the latter study, a small sample size (*n* = 31) could have contributed to the null results. None of these studies investigated the influence of high alexithymia on emotion recognition in faces wearing masks. Previous research on the effect of autistic traits shows that an extreme group approach can be important in detecting recognition impairments in the identification of emotions from masked faces [21]. Such studies could be important because impairments in the recognition of emotions from faces wearing face masks may become apparent only in high alexithymia. An interesting observation in this context is that alexithymia has been found to be associated with impairments in the Reading the Mind in the Eyes Test (RMET [48]), which assesses the ability to decode others’ emotional experiences based on images of the eye-region [49,50]. Impairments in recognizing the emotions of others from the eyes appear to characterize alexithymic men but not alexithymic women [51].

The eye tracking technique is an important tool to better understand attention allocation to emotional stimuli and facial features in face perception [52,53]. Eye tracking provides a rather direct measure of attention allocation, as the direction of eye gaze and focus of attention seem tightly coupled [54]. In general, people primarily examine the eyes when looking at facial expressions [55,56]. Eye tracking analyses revealed that during the identification of emotional facial expressions, attention to the eyes is about three times greater than to the mouth [57]. However, attention allocation to facial features varies to some extent as a function of emotion quality: the eyes seem to attract more attention in the case of angry and sad faces, whereas the mouth appears to receive more attention in the case of happy expressions [5,58,59]. There is evidence that increased attention to the eyes (compared to the mouth) is associated with a better identification of angry and sad facial expressions [60]. Yet, the recognition process of facial emotions cannot be reduced to simple feature processing since there is evidence of the involvement of holistic processing in the identification of facial emotional expressions [6,7]. Analyses of gaze behavior while viewing emotional facial expressions revealed sex differences in the orientation of visual attention to salient facial features: women look more at the eyes than men do, whereas males spend more time viewing the mouth [61,62]. It has been argued that the tendency of women to look more to the eyes may underlie the female advantage in facial emotion recognition [61].

Bird et al. [63] examined gaze behavior during the passive viewing of video clips in individuals with autism spectrum disorders. In this study, alexithymia predicted a reduction in participants’ attentional preference for the eyes when viewing actors in the videos. Fujiwara [64] investigated gaze patterns in participants by judging the mixture ratio of two emotional expressions blended into one face. Although alexithymic and non-alexithymic individuals were equally able to judge facial emotion blends, alexithymic individuals showed reduced attention to the eye region of emotional faces in comparison to non-alexithymic individuals. In alexithymic individuals, focusing attention on the eyes went along with diminished recognition of emotional expressions [64]. It appears that eye contact could be confusing or stressful for individuals with alexithymia. Similarly, during an RMET, alexithymic individuals dwelled less on the eye region than non-alexithymic controls although the groups did not differ with respect to emotion recognition accuracy and response times [65]. Thus, previous eye tracking results suggest that during passive viewing as well as during recognition of emotions from the eye region or faces, alexithymia seems to be related to reduced attention to the eyes. The eye avoidance appears not to be linked to impairments in emotion recognition. In a recent electrophysiological study, evidence of abnormalities in eye region processing of emotional expressions was reported in individuals with high levels of alexithymia [66]: alexithymic individuals relied less on perceptual processing of the eye region and exhibited diminished affective encoding for the eye region compared to non-alexithymic individuals.

The present study had two main objectives. The first was to explore the recognition of emotions in faces with a face mask in high-alexithymic compared to non-alexithymic individuals. We examined accuracy as well as speed of emotion recognition in faces with face masks. As the duration of facial stimulus presentation was rather long in our study (2 s), we expected that alexithymia would primarily have an adverse effect on response latencies. In our emotion recognition task, four different categories of facial expressions were shown, i.e., happy, fearful, disgusted, and neutral faces. Our second objective was to investigate the effect of alexithymia on attentional preference for the eyes when looking at emotional faces wearing face masks. To this aim, we used an eye-to-mask ratio based on fixation duration indicating attention allocation toward the eyes in comparison with the face mask, which covered the lower part of the face (including the mouth). It was hypothesized that alexithymic participants manifest less attentional preference for the eyes than non-alexithymic participants do. As previous research findings suggest that women are better at recognizing facial emotions [12,13] and look longer at the eyes during emotion recognition than men [61], we included the factor sex in our analyses of the behavioral and eye-tracking data. We assessed the depressed mood, state, and trait anxiety of participants because these negative effect variables had been found to go along with alexithymia [67,68]. Moreover, depressed mood and anxiety can have an impact on facial emotion recognition [69,70]. Verbal intelligence is an additional factor influencing labeling and recognition of emotions in facial expressions [71], which can explain relationships between alexithymia and emotion recognition [72]. Against this background, we decided to assess and, if necessary, control participants’ verbal abilities and affectivity. As we measured the speed of emotion recognition using manual keypress responses, we decided to assess participants’ visuomotor processing speed in a task with non-emotional stimuli. To this aim, we administered the Trail Making Test Part B [73].

## 2. Materials and Methods

### 2.1. Participants

Study participants were recruited through advertisements on online platforms and public places. The final sample consisted of 89 healthy individuals with a mean age of 24.37 years (SD: 4.67, range: 18–35). Interested individuals were interviewed via phone by doctoral medical students (trained and supervised by experienced clinical psychologists) to check inclusion and exclusion criteria. The presence of a diagnosed mental or neurological disorder, use of psychotropic medication, and visual impairments were exclusion criteria. Individuals undergoing psychiatric or psychotherapeutic treatments were excluded from the study. Persons matching these criteria were invited to fill out the 20-item Toronto Alexithymia Scale (TAS-20 [74,75,76]). They were classified as alexithymic or non-alexithymic using the procedure suggested by Bagby and Taylor [77]: values greater than 60 define clinically relevant levels of alexithymia, whereas values lower than 52 indicate the absence of alexithymia. Individuals with values between 52 and 60 were excluded from the study. The final sample comprised 38 alexithymic individuals (18 women) and 51 non-alexithymic individuals (26 women). We calculated a priori power analysis with the program G*Power 3.1 [78] to determine the required sample size to detect group × repeated measure interactions (*F*-tests for alexithymic vs. non-alexithymic individuals (group) and four emotional qualities of facial expression (within-subject measure)). The required total sample size to detect a medium-size effect of f = 0.25 was 36 given an alpha error probability of 0.05, a power of 0.95 (with two groups and four measurements), a correlation between repeated measures of 0.50, and a non-sphericity correction of 1. We expected group differences of medium effect size for the eye-to-mask gaze ratio and emotion recognition performance. In her eye-tracking study, Fujiwara [64] compared high and low alexithymic individuals and observed a medium-sized effect of group on eye preference (*ηp*^2^ = 0.06). Parker et al. [32] assessed recognition of facial expressions of basic emotions as a function of alexithymia: low alexithymic individuals showed a recognition score of 209.37 (with an SD of 54.79) whereas high alexithymic individuals had a recognition score of 180.18 (with an SD of 56.29). The latter results suggest a medium-sized effect of group (*d* = 0.53) regarding recognition of facial emotions in unmasked faces. Such an estimate of statistical power seems rather conservative for our experiment based on masked faces since emotions are more difficult to identify in masked faces than in unmasked faces [79]. Demographic data for both groups are presented in Table 1. All participants had normal vision, as determined by a Snellen eye chart. Individuals received a fee of EUR 30 for study participation. At the beginning of the experimental session, all participants gave their informed written consent to participate in the study. The ethics committee at the Medical Faculty of the University of Leipzig approved the present study.

### 2.2. Questionnaires and Tests

The 20-item Toronto Alexithymia Scale (TAS-20) is a widely used self-report measure of alexithymia with replicated validity and reliability [74,80] (German version [76]). The TAS-20 assesses three alexithymia components: difficulties in identifying feelings and distinguishing them from bodily sensations of emotional arousal, difficulties in describing feelings to other people, and an externally oriented thinking style. The total alexithymia score is the sum of the responses to all 20 items, which are rated on a 5-point scale, with a range of possible values from 20 to 100. In our sample, Cronbach’s alpha was 0.80 for the TAS-20 sum score.

The Beck Depression Inventory is a multiple-choice self-report test (BDI-II [81]; German version [82]), which measures the severity of depressive symptoms occurring over the previous two weeks on a behavioral, emotional, cognitive, and somatic level. The BDI-II comprises 21 items with four answer options (corresponding to 0 to 3 points). Summing ratings of all items yields scores ranging from 0 to 63. In our sample, Cronbach’s alpha for the BDI-II was 0.81.

The State-Trait Anxiety Inventory (STAI [83]; German version [84]) is a self-report measure of state and trait anxiety. The STAI consists of 20 items, respectively, that are rated on a 4-point scale. The trait version of the STAI measures stable interindividual differences in anxiety proneness, in appraising situations as threatening and avoiding anxiety-provoking situations. The state version of the STAI assesses anxiety as a temporary emotional state. Cronbach’s alphas for the STAI trait and STAI state were 0.91 and 0.88 in the present sample.

The Multiple-choice vocabulary intelligence test (Mehrfachwahl-Wortschatz-Intelligenztest, MWT-B [85]) is a performance test measuring facets of general intelligence, specifically crystallized, and verbal intelligence. The MWT-B consists of 37 items and has no time limit. Each item comprises one real word and four pronounceable pseudo-words (artificial words). The actually existing words have to be recognized. The number of correctly identified words can be converted to IQ scores.

The Trail Making Test Part B (TMT-B [73]) was administered to assess cognitive flexibility and psychomotor functioning. In this test, participants are required to connect numbers and letters in ascending order with a pen. The TMT-B consists of 25 items. The time to complete the task is measured (in seconds).

### 2.3. Emotion Recognition Task: Stimuli and Procedure

Face stimuli consisted of 80 frontal color photographs of twenty young models (10 female and 10 male), chosen from the MPI FACES database [86]. Each model posed four different facial expressions (happy, fearful, disgusted, and neutral). The photos of the models were digitally edited by superimposing a mask on the original images (see Figure 1 for examples of masked faces). The mask resembled a light blue surgical face mask and was adapted to match the length and width of the respective face so that it covered the face from the upper nose downwards. All photos were presented on a white background. The display size of each face photo on the screen was 19.3 cm high × 15.4 cm wide.

At the beginning of the experiment, participants received instruction on how to perform the recognition task. They were told that they would see photos of faces expressing the emotions of happiness, disgust, or fear. Moreover, they were informed that some faces would have a neutral expression. Participants were instructed to identify the expression of each face and to respond primarily as accurately but also as fast as possible. In each block, trials were shown in an individual random sequence.

Each trial had the following routine: after the presentation of a central fixation cross for 1000 ms, a facial stimulus was presented for 2000 ms. After the appearance of the face, participants could label the expressed facial emotion by mouse click in a forced choice manner. They saw the response categories side by side at the bottom of a white screen in black letters and entered their answers using a mouse. Each trial ended with the participants’ responses. The intertrial interval had a duration of 1000 ms. During the experiment, participants were seated in a chair at about 65 cm in front of the screen in a quiet room shielded from sunlight and illuminated by ceiling lights.

Hit rates (i.e., percentages of correct identifications of facial expressions) and reaction times (RT) of correct responses were calculated for each facial expression condition and each study participant.

### 2.4. Eye-Tracking: Apparatus and Eye Movement Parameters

A 24-inch LED monitor was administered for stimulus presentation (resolution: 1920 × 1200, refresh rate: 60 Hz). A Tobii Pro Fusion eye-tracker, fixed to the bottom of the monitor, was used to collect gaze data. The Tobii Pro Fusion records eye movement data at speeds up to 250 Hz per second. A standard 9-point calibration procedure was conducted before the experimental task to map eye position to screen coordinates. Tobii Pro Lab software (version 1.207.44884 ×64) was used to program the experiment, present the stimuli and collect and analyze the eye movement data (Tobii Technology, Stockholm, Sweden). Statistical analysis of the eye-tracking data was based on the 2000 ms presentation period of the face stimuli. Two areas of interest (AOI) were defined for each face: the eye region and the surface of the face mask. The AOI for the eye region was created around the eyes and above the face mask (see for an example Figure 2).

First, the eye tracking parameter duration of fixation was calculated as an indicator of attention allocation. The duration of fixation represents the sum of durations from all fixations (in milliseconds) that hit a specific AOI during a trial. The duration of fixation was determined for each AOI (eyes and mask) and each trial and then averaged for each participant. Second, an eyes-to-mask gaze ratio was computed. Similar indices of eye preference have been used, for example, in the studies of Fujiwara [64] and Bird et al. [63]. The eye-to-mouth gaze ratio of Bird et al. [63] refers to the time spent fixating the eyes relative to the total time spent fixating the eyes and the mouth of a face. Similarly, the eyes-to-mask gaze ratio used in our investigation was computed by dividing the total fixation time on the eyes by the sum of the total fixation times on the eyes and the face mask. Eyes-to-mask ratios greater than 0.5 indicate an attentional preference toward the eye region compared to the face mask.

### 2.5. General Procedure

After the screening session, if suitable, individuals were invited to participate in the eye-tracking experiment. The experimental session was conducted individually in the eye-tracking laboratory of the Department of Psychosomatic Medicine at the University of Leipzig. At the beginning of the session, study participants underwent vision screening using a Snellen eye chart and were asked to report sociodemographic information. Then, participants performed the eye-tracking experiment. Subsequently, the above-mentioned psychological tests and questionnaires were administered in a fixed order: BDI-II, STAI (state version), MWT-B, STAI (trait version), and TMT-B.

### 2.6. Statistical Analyses

To examine psychological characteristics and socio-demographic data as a function of study group and sex, two-factor univariate ANOVAs and Chi^2^-tests for contingency tables were performed. We administered *t*-tests for independent samples to explore differences in the subscales of the TAS-20 between alexithymic women and alexithymic men. Hit rates, response latencies for correct responses, and the eye tracking parameters (eyes-to-mask gaze ratio, fixation duration on the eyes, and fixation duration on the face mask) were analyzed using 4 × 2 × 2 mixed ANOVAs with emotional quality of facial expression (happiness, fear, disgust, and neutral) as within-subjects factor and study group (alexithymia vs. non-alexithymia) and sex (women vs. men) as between-subjects factors. In case the assumption of sphericity was violated, we used the Greenhouse–Geisser method to correct degrees of freedom [87]. Bonferroni-corrected pairwise comparisons were calculated as a follow-up to analyze pairwise differences. Product–moment correlation analysis was used to examine the relationships between psychological characteristics (measures of affectivity, intelligence, and psychomotor functioning), hit rates, RTs, and eyes-to-mask gaze ratios. Results were considered significant at an alpha level of *p* ≤ 0.05 (two-tailed). SPSS 29.0 (IBM Corp., Armonk, NY, USA) was used to analyze the data.

## 3. Results

### 3.1. Sociodemographic and Psychological Variables

Descriptive statistics for sociodemographic data and psychological characteristics as a function of study group and sex are shown in Table 1. According to the results of *Chi*^2^ testing, there was no significant association between alexithymia (group) and sex, *Chi*^2^ (1) = 0.01, *p* = 0.93. There was also no significant relationship between level of school education and sex, *Chi*^2^ (2) = 5.42, *p* = 0.07. However, a significant association between the level of school education with alexithymia (group) was observed, *Chi*^2^ (2) = 7.11, *p* < 0.05. In the alexithymia group, five individuals had a low level of school education (<12 years), whereas in the non-alexithymia group, no participant had a low level of school education (see Table 1).

The ANOVA results for the TAS-20 score revealed a large effect of the study group, *F*(1, 85) = 463.63, *p* < 0.001, *ηp*^2^ = 0.84. The effect of sex and the interaction group × sex were non-significant. The ANOVA findings for age showed only an effect of sex, *F*(1, 85) = 6.01, *p* < 0.05, *ηp*^2^ = 0.07. In our study, male participants were older than female participants (see Table 1). The findings for the MWT-B indicated only a significant effect of group, *F*(1, 85) = 4.47, *p* < 0.05, *ηp*^2^ = 0.05. Alexithymic individuals had lower verbal intelligence scores than non-alexithymic individuals. For the TMT-B, no significant effects were found (all *p*s > 0.30). The ANOVA results for the BDI-II suggest only a significant effect of group *F*(1, 85) = 8.44, *p* < 0.01, *ηp*^2^ = 0.09. Alexithymic individuals reported more depressive symptoms compared to non-alexithymic individuals (see Table 1). For the state version of the STAI, we observed no significant effects (all *p*s > 0.10). For the trait version of the STAI, significant effects of group, *F*(1, 85) = 17.63, *p* < 0.001, *ηp*^2^ = 0.17, and sex, *F*(1, 85) = 4.67, *p* < 0.05, *ηp*^2^ = 0.05, were revealed. The interaction group × sex was not significant. Alexithymic individuals were found to describe themselves as more trait anxious than non-alexithymic individuals and women described themselves as more trait anxious than men in our study (see Table 1).

We explored whether alexithymic women differed from alexithymic men on the alexithymia subscales of the TAS-20. Results from *t*-tests indicated that alexithymic women had higher scores on the subscale difficulties in identifying feelings (26.56) than alexithymic men (22.65), *t*(36) = 3.31, *p* < 0.01, *d* = 1.07. In our study, alexithymic men had higher externally-oriented thinking scores (23.60) than alexithymic women (20.72), *t*(36) = −2.36, *p* < 0.05, *d* = 0.77. No differences between alexithymic women and men were found for the subscale difficulties in describing feelings (20.83 vs. 19.90)

### 3.2. Emotion Recognition Performance

Mean hit rates as a function of emotional quality of facial expression, group, and sex are presented in Table 2. A 4 × 2 × 2 mixed ANOVA on hit rates yielded only a main effect of emotional quality of facial expression, *F*(2.46, 209.51) = 8.80, *p* < 0.001, *ηp*^2^ = 0.09. No other significant effects were observed (all *p*s > 0.10). According to Bonferroni-adjusted pairwise comparisons, the hit rate was higher for fearful faces (0.910) compared to neutral faces (0.819), *p* < 0.001, and disgusted faces (0.868), *p* ≤ 0.05. Moreover, the hit rate was higher for happy faces (0.875) than for neutral faces, *p* ≤ 0.05. Hit rates did not differ between the happy and the disgusted face condition, between the fearful and the happy face condition, and between the neutral and the disgust condition.

Mean reaction latencies for correct responses as a function of emotional quality of facial expression, group, and sex are displayed in Table 3. A 4 × 2 × 2 mixed ANOVA based on reaction latencies yielded only a main effect of emotional quality of facial expression, *F*(2.52, 213.95) = 24.10, *p* < 0.001, *ηp*^2^ = 0.22. No other significant effects were observed (all *p*s > 0.10). Bonferroni-adjusted pairwise comparisons showed that RTs were lower for neutral faces (3362 ms) compared to disgusted faces (3887 ms), *p* < 0.001, and fear faces (3620 ms), *p* < 0.001. RTs were also lower for happy faces (3332 ms) compared to disgusted faces, *p* < 0.001, and fear faces, *p* < 0.01. Response latencies did not differ from each other for neutral and happy faces. Finally, response speed was faster for fearful than for disgusted faces, *p* < 0.01.

### 3.3. Eyes-to-Mask Gaze Ratio

Mean eyes-to-mask gaze ratios depending on the emotional quality of facial expressions, group, and sex are reported in Table 4. A 4 × 2 × 2 mixed ANOVA on eyes-to-mask gaze ratios revealed a main effect of emotional quality, *F*(2.23, 195.36) = 19.31, *p* < 0.001, *ηp*^2^ = 0.18, and an interaction group × sex, *F*(1, 85) = 7.57, *p* < 0.01, *ηp*^2^ = 0.08. No other significant effects were found (all *p*s > 0.25). Bonferroni-corrected pairwise comparisons of eyes-to-mask gaze ratios showed that attentional preference of the eyes relative to the mask was smaller for neutral (0.898) than for happy (0.911), disgusted (0.925), and fear faces (0.918), *p*s < 0.05. Moreover, the attentional preference of the eyes relative to the mask was greater for disgusted faces compared to happy faces (*p* < 0.001). No other significant differences between face conditions were observed for the eyes-to-mask gaze ratio.

To further analyze the group × sex interaction, separate ANOVAs on overall eyes-to-mask gaze ratios with the factor group were conducted for women and men. For women, a marginally significant effect of group was observed, *F*(1, 42) = 3.48, *p* = 0.07, *ηp*^2^ = 0.08. Attentional preference of the eyes relative to the mask tended to be greater in alexithymic women (0.933) compared to non-alexithymic women (0.907) (see Figure 3). For men, the effect of the group was significant, *F*(1, 43) = 4.30, *p* < 0.05, *ηp*^2^ = 0.09. Non-alexithymic men exhibited more attentional preference for the eyes relative to the mask (0.926) than alexithymic men (0.886) (see Figure 3).

To better understand the observed alexithymia-related effects for the eyes-to-mask gaze ratio, we calculated additional ANOVAs for the parameters fixation duration on the eyes and fixation duration on the mask (see below).

### 3.4. Correlations of Affectivity, Intelligence, and Psychomotor Functioning with Recognition Performance and Eyes-to-Mask Gaze Ratio

The results of product–moment correlations between psychological measures, emotion recognition, and eyes-to-mask gaze ratio are shown in Table 5. Only intelligence (but not psychomotor functioning, depression, or anxiety) was correlated with the overall hit rate in the emotion recognition task (see Table 5). High intelligence went along with more correct identifications of facial expressions. Psychomotor functioning was found to be positively correlated with RT for correct responses in the emotion recognition task (see Table 5). Thus, slow psychomotor performance in the TMT-B was associated with longer response latencies in emotion recognition. Intelligence, depression, and anxiety were not correlated with RT in the emotion recognition task. Finally, there were no correlations between psychological measures and overall eyes-to-mask gaze ratio (see Table 5).

### 3.5. Correlations between Recognition Performance and Eyes-to-Mask Gaze Ratio

The results of product–moment correlations between emotion recognition and eyes-to-mask gaze ratio are shown in Table 6 for the whole sample and the alexithymic and non-alexithymic subsamples. In none of the samples, correlations were found between emotion recognition (hit rate and response latency of correct responses) and the eyes-to-mask gaze ratio. Thus, there was no evidence of a relation of hit rates or response times with an attentional preference of the eyes relative to the mask. In the whole sample and the non-alexithymic subsample, significant negative correlations were observed between hit rate and response time (see Table 6 for details). This indicates that high hit rates were associated with fast responses in our emotion recognition task. In the alexithymic subsample, the correlation between hit rate and response latency of correct responses was also negative but non-significant (*r* = −0.21).

### 3.6. Fixation Duration on the Eyes

Mean fixation times on the eyes depending on the emotional quality of facial expressions, group, and sex are shown in Table 7. A 4 × 2 × 2 mixed ANOVA on fixation duration on the eyes yielded no significant main or interaction effects (all *p*s > 0.10).

### 3.7. Fixation Duration on the Face Mask

Mean fixation durations on the face mask as a function of the emotional quality of facial expressions, group, and sex are presented in Table 8. A 4 × 2 × 2 mixed ANOVA on fixation times on the face mask revealed a significant effect of emotional quality, *F*(2.14, 182.33) = 23.04, *p* < 0.001, *ηp*^2^ = 0.21, and a significant interaction group × sex, *F*(1, 85) = 9.97, *p* < 0.005, *ηp*^2^ = 0.10. No other significant effects were found (all *p*s > 0.10). To further analyze the interaction group × sex, ANOVAs on overall fixation duration on the mask with the factor group were performed for women and men separately. For women, no significant effect of group was observed, *F*(1, 42) = 2.43, *p* = 0.13, *ηp*^2^ = 0.05. Fixation time on the face mask of alexithymic women (100 ms) did not differ from that of non-alexithymic women (131 ms). For men, the effect of the group was significant, *F*(1, 43) = 7.80, *p* < 0.01, *ηp*^2^ = 0.15. Alexithymic men fixated on the mask longer (173 ms) than non-alexithymic men (99 ms).

## 4. Discussion

Based on emotional expressions, observers can infer information about other persons’ behavioral intentions, attitudes, and relational orientation [88]. The ability to correctly interpret other people’s facial emotions is crucial in subsequently deciding on appropriate actions [89]. The present study investigated the recognition of emotions in faces with a face mask and attentional preference for the eyes during emotion recognition in high-alexithymic compared to non-alexithymic individuals. We included the factor of biological sex in our data analyses since women have been found to better recognize facial emotions [12,13] and to look longer at the eyes during emotion recognition than men [61]. During the COVID-19 pandemic, face masks were an important protective measure to decrease the spread of the coronavirus but it has been shown that face masks have detrimental effects on emotion identification [16,17,18]. It is important to clarify whether personality traits known to influence emotion perception such as alexithymia could lead to particularly pronounced impairments in emotion recognition from faces wearing masks. Beyond the pandemic, such knowledge could be of significance for clinicians who use face masks in their everyday professional lives and treat alexithymic patients. Doctor–patient or nurse–patient relations, in general, require fast and correct interpretation of emotional states for better patient outcomes.

Our data do not confirm the hypothesis that high-alexithymic individuals manifest impairments in the recognition of emotions in faces with face masks. In our study, high-alexithymic individuals did not differ from non-alexithymic individuals in accuracy or speed of emotion recognition for masked faces. Thus, contrary to expectation, high-alexithymic participants were as fast and accurate as non-alexithymic participants in the identification of masked happy, fearful, disgusted, and neutral facial expressions. The present results are in line with and expand previous findings of correlational emotion recognition studies based on masked facial expressions, which indicated no associations of alexithymia with recognition accuracy [46] or speed of emotion identification [47]. Our findings suggest that even high levels of alexithymia might not have adverse effects on the efficiency of emotion recognition from faces wearing masks. The present data are also in line with the findings of Carbon et al. [22], who found no effect on participants’ emotional intelligence regarding their accuracy in recognizing emotional states in masked faces. However, before strong conclusions can be drawn, further research in this area is needed. It can be criticized that our task was not difficult enough to detect the effects of alexithymia on identification performance. In our experiment, only intense emotional expressions were administered as face stimuli. It is possible that emotional facial expressions of low intensity could be more suitable to reveal the effects of alexithymia on emotion recognition in masked faces. The present null results are somewhat surprising considering that previous studies observed rather consistent evidence for emotion identification impairments in alexithymia when the presentation of facial stimuli was suboptimal. That is, when faces were shown in degraded quality or with temporal constraints [35,36,37,38] or when only the eye region of facial expressions was presented [49,50,51].

According to our data, participants’ sex had no effect on the accuracy and speed of emotion recognition. The highest hit rate in our study was observed for masked fearful faces, which were better recognized than masked disgusted faces. This finding is in accordance with previous research results indicating that face masks reduce recognition performance to a varying extent across different emotion qualities. The detrimental effects of face masks on emotion recognition seem to be large for disgust [20,79]. In contrast, face masks appear to have no or only a little effect on the identification of fear [19,90,91]. Key features of facial fear expressions are raised eyebrows and eyelids as well as enlarged eye-whites [92]. The eye region seems to be a particularly important facial feature for recognizing fear [93] and is not covered by face masks.

To evaluate the difficulty of our emotion recognition task we compared our overall hit rates (averaged across study groups) with hit rates of other investigations that also examined emotion recognition based on the MPI faces database [22,86] in samples of young and healthy individuals using unmasked or masked facial expressions. Our hit rates for masked happy faces (0.87) and masked neutral faces (0.82) were lower than those reported by Ebner et al. [86] for unmasked young happy faces (0.98) and unmasked young neutral faces (0.93), averaged across young female, and male raters. Our hit rates for masked fearful faces (0.91) and masked disgusted faces (0.87) were higher than those reported by Ebner et al. [86] for unmasked young fearful faces (0.85) and unmasked young disgusted faces (0.79), averaged across young female and male raters. Thus, for fearful and disgusted expressions, it can be noted that adding a face mask might have had a recognition-enhancing effect. At least for fearful faces, this effect could be due to the mask directing participants’ attention to a diagnostically highly relevant area, the eye region. When comparing our hit rates with those of Carbon et al. [22], who also investigated emotion recognition in faces with masks, similar hit rates were found for fearful faces (0.91 vs. 0.93). Our hit rates were higher for happy faces (0.87 vs. 0.75) and in particular for disgusted faces (0.87 vs. 0.40) than those of Carbon et al. [22]. Finally, our hit rate for neutral faces was lower than that reported by Carbon et al. (0.82 vs. 0.95). All in all, it appears that, in general, our emotion recognition task could be less difficult than the task applied by Carbon et al. [22]. Even though both studies administered MPI faces our task required only the differentiation between four categories of facial expressions, whereas the task of Carbon et al. [22] required the differentiation between six categories of facial expressions. Moreover, it appears that the face masks applied in the study of Carbon et al. [22] covered more area on the upper nose and the lower eye region than the face masks in our study. These differences in face masking could have led to less recognition of disgust and happiness in the study of Carbon et al. [22].

The main objective of our study was to examine the effect of alexithymia on attentional preference for the eyes when looking at emotional faces wearing face masks. The present results confirmed, at least in part, our expectation that alexithymic participants manifest less attentional preference for the eyes than non-alexithymic participants do. Alexithymic men showed less attention allocation toward the eyes in comparison with the face mask compared to non-alexithymic men. In contrast, there were no differences in attentional preference for the eyes between alexithymic and non-alexithymic women. It is noteworthy that, on a descriptive level, alexithymic women had even higher attentional preference scores for the eyes than non-alexithymic women. Thus, it appears that biological sex could be an important variable to consider when studying the relationships between alexithymia and attentional preference for the eyes. According to the results of our additional separate analyses concerning the two AOIs eyes and face mask, alexithymic men did not differ from non-alexithymic men in sustained attention on the eyes but in sustained attention on the face mask. That is, alexithymic men fixated longer on the face mask than non-alexithymic ones. This result pattern indicates that subgroup differences in the eye-preference ratio were driven by increased face mask-viewing in alexithymic men. Overall, in our study, the fixation duration on the eyes was more than 10 times greater in comparison with the fixation duration on the face mask. Previous research based on unmasked faces has shown that attention to the eyes is about three times greater than to the mouth during the identification of emotional facial expressions [57]. It is known that people move their gaze preferentially to the eyes of faces in which the lower part is hidden by a scarf or mask [94]. In our study, face masks likely attracted only very little attention (compared to the eyes) since they consisted of a poorly contoured surface containing little visual information.

In our study, visual attention to the eyes was not related to a better or faster recognition of facial emotions—neither in the whole sample nor in the alexithymic or non-alexithymic subsamples. It appears that participants did not benefit from longer fixation on the eye region (relative to the mask) concerning accuracy or speed of emotion identification in facial expressions. A possible explanation is that ceiling effects in the current study (low difficulty of our emotion recognition task) limited the ability to see a positive relationship between visual attention to the eyes and emotion recognition performance.

Prima facie, our eye tracking results for alexithymic men seem to confirm the finding of Fujiwara [64] who examined gaze patterns in participants judging the mixture ratio of two emotional expressions blended into one unmasked face. Fujiwara [64] observed that alexithymic individuals manifested a reduced eyes-to-mouth gaze ratio in comparison with non-alexithymic individuals. However, separate analyses for dwell time on the eyes and the mouth revealed that the alexithymia group showed significantly less eye-viewing but not more mouth-viewing [64]. It was concluded that the finding of group differences in the eye-preference ratio was mainly driven by reduced eye-viewing in alexithymic individuals [64]. This pattern of results contrasts with our findings, which indicates no differences in sustained attention on the eyes but more sustained attention on the lower parts of the face, i.e., the face mask, in alexithymic compared to non-alexithymic men. All in all, our null results for gaze duration on the eyes and alexithymia differ not only from Fujiwara’s findings [64] but also from those of Zimmermann et al. [65] who observed that during an RMET, alexithymic individuals dwelled less on the eyes than non-alexithymic controls. The latter difference in results could be due to the different types of facial stimuli used: in the RMET, only images of the eye region were presented (with averted gaze or gaze directed at the observer), whereas, in our recognition task, images of whole faces with direct gaze wearing a face mask were displayed. Moreover, the participants in Zimmermann et al.’s neuroimaging study [65] were lying in a scanner during the RMET (a rather stressful situation) whereas our participants sat upright in front of a monitor in a quiet room. Based on our data, it can be concluded that in case the lower part of the face is covered by a mask and participants’ gaze appears strongly directed toward the eyes, alexithymia does not have an effect on visual attention to others’ eyes. Thus, we found no evidence for an alexithymia-related eye avoidance for faces with face masks.

It is important to note that in our study, alexithymic men looked longer at the face mask than non-alexithymic men. That means that during emotion recognition, alexithymic men oriented their attention more to an area of the face, which contains little visual information and is task-irrelevant, i.e., not informative concerning the other person’s emotional state. As mentioned above, alexithymic women did not differ from non-alexithymic women in their visual attention directed to the eyes or the lower face region. Interestingly, in our study, we did not find a general effect of sex on attentional preference even though, according to previous research, women look longer at the eyes during emotion recognition than men [61]. The present results suggest that future alexithymia research on gaze behavior on faces may consider sex as a moderating variable. Importantly, in our study, alexithymic men differed from alexithymic women on two alexithymia facets: they were characterized by a more pronounced externally oriented thinking style and less difficulty identifying emotions. The alexithymia facet of externally oriented thinking has been reported to be associated with difficulties in interpersonal relations. Individuals high in externally oriented thinking appear to be characterized by low empathic concern and high self-centeredness in interpersonal relationships [95,96]. So far, little research has been conducted on the phenomenology and etiology of sex differences in alexithymia. Levant [97,98] proposed the normative male alexithymia hypothesis to account for a socialized pattern of restrictive emotionality in men influenced by traditional masculinity ideology. According to this approach, traditionally reared men restrict the expression of emotions, which signal vulnerability and fragility such as fear and sadness or attachment emotions such as affection [99]. Normative male alexithymia has been shown to be related to poor communication quality and fear of intimacy [100]. It is an interesting question as to whether heightened attention orientation to facial areas containing no information on the others’ emotional state could be part of a suboptimal gaze characteristic of alexithymic men during interpersonal contact.

Several limitations of the present investigation should be mentioned. First, the generalizability of our findings is restricted by the fact that the majority of study participants were well-educated young individuals. It is also a limitation of the present study that no data on emotion recognition in unmasked facial expressions are available. These data would have allowed group comparisons in emotion recognition across masked and unmasked face conditions. It can be criticized that we did not assess autistic traits in our study participants, as previous research suggests that these traits can be related to poor emotion recognition in faces [101] as well as atypical scanning behavior in response to social stimuli [102]. A further limitation of our study is the sole reliance on self-reporting for assessing alexithymia. Doubts have been raised concerning the validity of self-report questionnaires assessing alexithymia, as such instruments seem to depend on the ability to attend to and describe one’s feelings correctly [103]. However, over the last decades, research has provided considerable evidence for the reliability and validity of the TAS-20 [80]. It can also be criticized that our masked faces represent artificial facial stimuli that may not have ecological validity. In the present study, images of light blue face masks with ear loops were superimposed over the original face stimuli by using image processing software. Interestingly, a recent investigation compared the accuracy of emotion recognition from graphically manipulated facial stimuli and stimuli where the emotions were posed by people wearing masks [104]. According to these findings, the effects on emotion recognition were generally similar for stimuli where the emotions were posed by people wearing masks and stimuli where emotions were posed without masks and masks subsequently added by graphics software.

## 5. Conclusions

According to our results, high levels of alexithymia may not have adverse effects on the efficiency of emotion recognition from faces wearing masks. High alexithymic individuals identify the emotions of fear, disgust, and happiness in masked facial expressions just as well and fast as non-alexithymic individuals. Our data suggest that even when participants’ gaze is strongly directed toward the eyes as in our experiment, alexithymia does not affect visual attention to others’ eyes. Thus, we found no evidence for an alexithymia-related eye avoidance for faces with face masks. However, during emotion recognition, alexithymic men seem to look longer at the face mask than non-alexithymic men, which is an area containing no information on the emotional state of the person. Against this background, it appears advisable that future research on gaze behavior during facial emotion recognition in alexithymia considers sex as a moderating variable.

## Figures and Tables

**Figure 1 behavsci-14-00343-f001:**
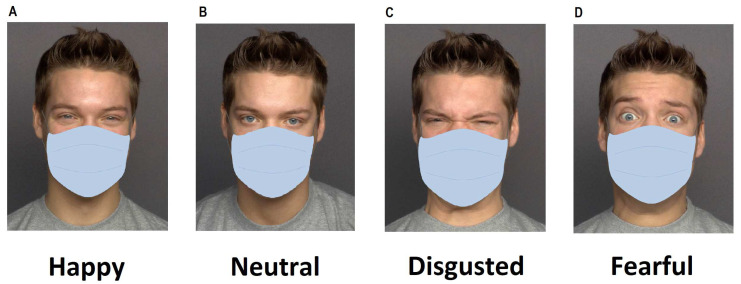
Examples of facial stimuli administered in the emotion recognition task: (**A**) happy expression, (**B**) neutral expression, (**C**) disgusted expression, and (**D**) fearful expression. The original images were taken from the MPI FACES database [86]. The depicted face is 066_y_m.

**Figure 2 behavsci-14-00343-f002:**
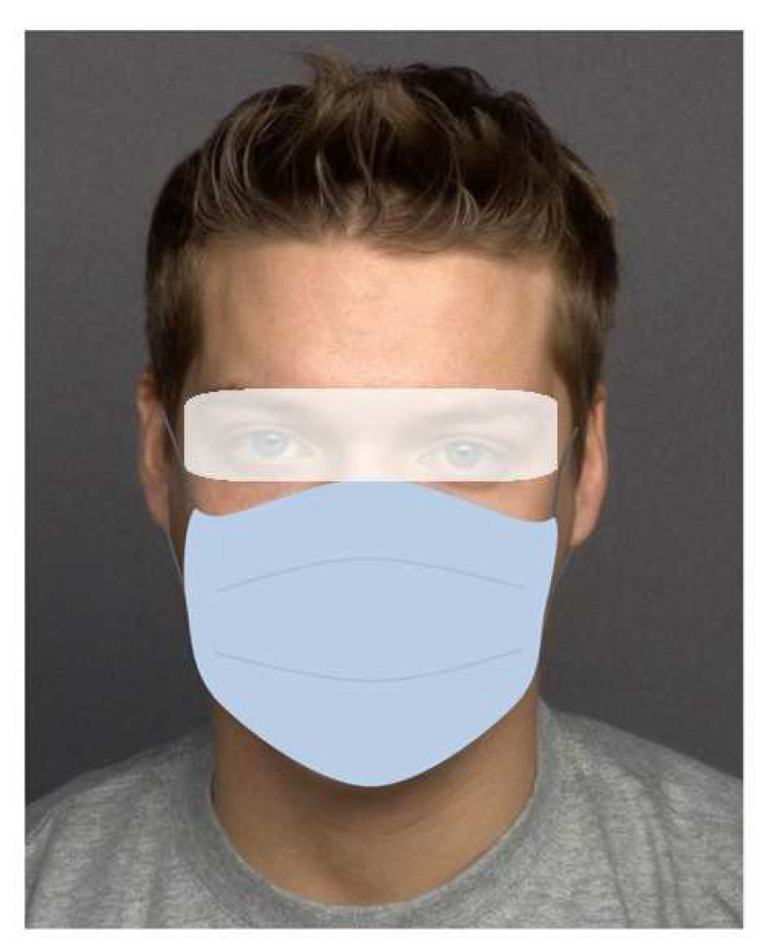
Areas of interest in our eye tracking experiment: eyes (white coloring) and face mask (blue coloring). The depicted face shows model 066_y_m from the MPI FACES database [86].

**Figure 3 behavsci-14-00343-f003:**
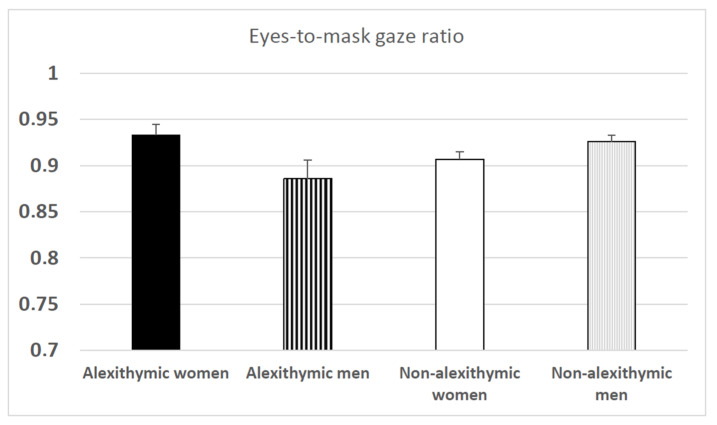
Overall eyes-to-mask gaze ratio as a function of alexithymia and sex (means with standard errors).

**Table 1 behavsci-14-00343-t001:** Demographic and psychological test data as a function of alexithymia and sex (means with standard deviations (in brackets)).

Variable	Alexithymic Women (*n* = 18)	Alexithymic Men (*n* = 20)	Non-Alexithymic Women (*n* = 26)	Non-Alexithymic Men (*n* = 25)
TAS-20 (sum score)	68.11 (4.30) ^A^	66.15 (5.00) ^A^	39.12 (7.15) ^B^	38.72 (6.79) ^B^
Age (years)	23.61 (4.77) ^A,B^	25.55 (5.08) ^A,B^	22.77 (4.32) ^B^	25.64 (4.25) ^A^
Level of				
school education				
N 10th grade	0	4	0	0
N 11th grade	0	1	0	0
N 12th grade	18	15	26	25
MWT-B (IQ)	105.50 (8.62) ^A^	108.00 (10.64) ^A,B^	109.81 (12.35) ^A,B^	114.00 (12.49) ^B^
TMT-B (seconds)	56.44 (17.17) ^A^	56.75 (12.26) ^A^	54.35 (16.22) ^A^	52.56 (10.10) ^A^
BDI-II (sum score)	11.17 (6.13) ^A^	9.40 (6.19) ^A,B^	7.62 (5.17) ^B,C^	6.08 (4.79) ^C^
STAI-S (item score)	1.99 (0.53) ^A^	1.83 (0.41) ^A^	1.78 (0.36) ^A^	1.76 (0.29) ^A^
STAI-T (item score)	2.36 (0.47) ^A^	2.02 (0.47) ^B^	1.83 (0.48) ^B,C^	1.76 (0.33) ^C^

^ABC^ Within a row means without a common superscript differ (independent *t*-tests, *p* < 0.05); TAS-20 = 20-Item Toronto-Alexithymia Scale; MWT-B = Multiple-choice vocabulary test version B; TMT-B = Trail-Making-Test, version B; BDI-II = Beck Depression Inventory-II; STAI-S = State-Trait Anxiety Inventory—state version; STAI-T = State-Trait Anxiety Inventory—trait version.

**Table 2 behavsci-14-00343-t002:** Hit rate for facial expression conditions as a function of alexithymia and sex (means with standard deviations (in brackets)).

Expression Condition	Alexithymic Women (*n* = 18)	Alexithymic Men (*n* = 20)	Non-Alexithymic Women (*n* = 26)	Non-Alexithymic Men (*n* = 25)
Happiness	0.880 (0.069)	0.814 (0.153)	0.907 (0.076)	0.899 (0.077)
Disgust	0.822 (0.169)	0.880 (0.145)	0.886 (0.171)	0.884 (0.120)
Fear	0.880 (0.139)	0.927 (0.079)	0.936 (0.082)	0.898 (0.118)
Neutral	0.843 (0.086)	0.794 (0.152)	0.844 (0.143)	0.793 (0.178)

**Table 3 behavsci-14-00343-t003:** Response latencies of correct responses (in ms) for facial expression conditions as a function of alexithymia and sex (means with standard deviations (in brackets)).

Expression Condition	Alexithymic Women (*n* = 18)	Alexithymic Men (*n* = 20)	Non-Alexithymic Women (*n* = 26)	Non-Alexithymic Men (*n* = 25)
Happiness	3270 (399)	3454 (612)	3324 (532)	3281 (637)
Disgust	3843 (669)	3905 (663)	3762 (756)	4036 (757)
Fear	3471 (498)	3910 (754)	3511 (544)	3586 (738)
Neutral	3263 (436)	3412 (496)	3374 (444)	3399 (484)

**Table 4 behavsci-14-00343-t004:** Preference ratio eyes-to-mask for facial expression conditions as a function of alexithymia and sex (means with standard deviations (in brackets)).

Expression Condition	Alexithymic Women (*n* = 18)	Alexithymic Men (*n* = 20)	Non-Alexithymic Women (*n* = 26)	Non-Alexithymic Men (*n* = 25)
Happiness	0.936 (0.046)	0.881 (0.095)	0.902 (0.048)	0.924 (0.043)
Disgust	0.943 (0.060)	0.894 (0.096)	0.921 (0.035)	0.943 (0.037)
Fear	0.937 (0.044)	0.892 (0.096)	0.914 (0.032)	0.930 (0.033)
Neutral	0.914 (0.071)	0.876 (0.080)	0.892 (0.060)	0.908 (0.046)

**Table 5 behavsci-14-00343-t005:** Product–moment correlations of psychological measures with overall hit rate, overall RT for correct responses, and overall eyes-to-mask gaze ratio in the emotion recognition task (*n* = 89).

Variable	Hit Rate	Response Latency	Eyes-to-Mask Gaze Ratio
MWT-B (IQ)	0.24 *	−0.15	0.07
TMT-B	−0.19	0.30 **	−0.09
BDI-II	0.07	−0.05	−0.07
STAI-S	0.11	−0.05	0.07
STAI-T	0.06	−0.06	0.00

MWT-B = Multiple-choice vocabulary test version B; TMT-B = Trail-Making-Test, version B; BDI-II = Beck Depression Inventory-II; STAI-S = State-Trait Anxiety Inventory—state version; STAI-T = State-Trait Anxiety Inventory—trait version. * *p* < 0.05; ** *p* < 0.01 (two-tailed).

**Table 6 behavsci-14-00343-t006:** Product–moment correlations between overall hit rate, overall RT for correct responses, and overall eyes-to-mask gaze ratio in the whole sample (*n* = 89), the alexithymic sample (*n* = 38), and the non-alexithymic sample (*n* = 51).

Variable	Response Latency	Eyes-to-Mask Ratio	Mean (SD)
** Whole sample: **			
Hit rate	−0.33 *	0.01	0.87 (0.08)
Response latency		0.04	3550 (456)
Eyes-to-mask ratio			0.91 (0.06)
** Alexithymic sample: **			
Hit rate	−0.21	0.09	0.85 (0.07)
Response latency		−0.10	3572 (425)
Eyes-to-mask ratio			0.91 (0.08)
** Non-alexithymic sample: **			
Hit rate	−0.41 *	−0.14	0.88 (0.08)
Response latency		0.24	3533 (482)
Eyes-to-mask ratio			0.92 (0.04)

* *p* < 0.01 (two-tailed).

**Table 7 behavsci-14-00343-t007:** Fixation duration on the eyes (in ms) for facial expression conditions as a function of alexithymia and sex (means with standard deviations (in brackets)).

Expression Condition	Alexithymic Women (*n* = 18)	Alexithymic Men (*n* = 20)	Non-Alexithymic Women (*n* = 26)	Non-Alexithymic Men (*n* = 25)
Happiness	1449 (218)	1388 (320)	1310 (278)	1325 (390)
Disgust	1454 (247)	1390 (331)	1292 (274)	1369 (402)
Fear	1429 (236)	1395 (323)	1320 (276)	1335 (390)
Neutral	1420 (241)	1374 (298)	1325 (281)	1310 (406)

**Table 8 behavsci-14-00343-t008:** Fixation duration on the mask (in ms) for facial expression conditions as a function of alexithymia and sex (means with standard deviations (in brackets)).

Expression Condition	Alexithymic Women (*n* = 18)	Alexithymic Men (*n* = 20)	Non-Alexithymic Women (*n* = 26)	Non-Alexithymic Men (*n* = 25)
Happiness	97 (67)	179 (129)	137 (63)	100 (56)
Disgust	84 (85)	160 (128)	107 (46)	78 (51)
Fear	91 (62)	162 (128)	122 (48)	94 (48)
Neutral	130 (106)	191 (112)	157 (89)	123 (64)

## Data Availability

The datasets used and analyzed during the current study are available from the corresponding author on reasonable request.

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
