# Peer review of "Recognizing and Looking at Masked Emotional Faces in Alexithymia"

_behavsci, 2024, doi:10.3390/bs14040343_

Round 1
Reviewer 1 Report
Comments and Suggestions for Authors
Your research question was very interesting and I was looking forward to reading your paper. You collected some interesting data, but made a number of choices that left me wondering about why you made them. You may have a good story here, but so far you have yet to tell it. I'll note that my issue is not with the findings and you may address my concerns and still have the same findings, but the concerns are more about the relationship between your variables, your stimulus, and your conclusions.
In order in which they occurred to me (sometimes later in the paper after reading something else)
- Null hypotheses. The big finding you report are null hypotheses, and I fine with that given your power analysis, though I'd appreciate a discussion of why you would make the estimate of a medium effect size. Do you have an example where you found such an effect size in this area of the literature? Further, in looking at your literature review, you note the importance of power on one null effect, but you actually report several null effects(e.g. refs 20 & 21). Given your finding and these findings, you can make a better estimate of effect size and you can calculate the power on the other null effects. Such an effort would certainly solidify the context of your findings and truly inform you if you had enough power.
- Why did you choose the emotions that you chose? You reviewed differences relevant to sadness and anger and the use of the eyes, but then you chose disgust and fearful. You accented the difference in cue location and cue utilization, but then didn't back up your choice with a rationale.
- I was surprised that you did not include an unmasked condition. Even just a small subset to balance out the other findings showing the same group differences reported in literature would have been very helpful. I don't think it would have taken too much time and given repeated measures, it would not have greatly hampered power. You could also explored the possibility that the full face might be more confusing since it offers too many sources of information instead of the masked condition which leaves only a smaller area of focus.
- Alternatively, since the unmasked faces were of the standardized MPI faces you could have seen whether there was a difference from each group to the normed data of that data set. Some sort of baseline would also help your study assert what it to claim.
- Using Ekman's AU system with the prototypical faces, what AUs are still visible in the masked presentation mode? Is there a possibility that the eyes and brows are more diagnostic versus full set of available cues with the mask on across the emotion you selected? The forced choice task, which is fine for the purposes of this study, really requires identifying the relevant cues and the mask narrows the focus of cues. It's possible you are increasing the diagnosticity of the remaining cues - which is also why a baseline to unmasked faces would be helpful.
- Any table that shows group differences should have subscript that identify which groups are different.
- What are the correlations between DVs? A relationship between DVs is an opportunity to increase power and reduce the number of comparisons between IVs and DVs, the primary concern in Type I error. Can you consolidate? Regardless a table of correlations between IVs with each other and DVs with each other are very helpful.
- As I am looking at your literature review, your results, and some aspects of your discussion, it seems to me that there might be more than one path to alexithymia. One is through autism, an other might be through lack of verbal facility to label emotions. There may be others. I am particularly concerned about autism. Your exclusion criterion suggest no neurological disorder (which should mean no autism), but given your literature review should explicitly say including no autism. Further, did you screen with some quick screen tools that are now available. Autism is still under diagnosed and I would think you'd want to consider a score on a screening tool as a possible covariate - even if to quickly rule out.
- I want to highlight that as you have described it, Alexithymia is issue in the production and self-labeling of emotion and emotion recognition is the ability to see it in others. What is the mechanism by which these two processes are connected? That connection will speak to what you may or may not have found.
- line 47 mentions "personality difference in emotion recognition." I have not heard it labeled as such. I think you are safer with individual difference.
Author Response
REVIEWER 1
Your research question was very interesting and I was looking forward to reading your paper. You collected some interesting data, but made a number of choices that left me wondering about why you made them. You may have a good story here, but so far you have yet to tell it. I'll note that my issue is not with the findings and you may address my concerns and still have the same findings, but the concerns are more about the relationship between your variables, your stimulus, and your conclusions.
In order in which they occurred to me (sometimes later in the paper after reading something else)
- Null hypotheses. The big finding you report are null hypotheses, and I fine with that given your power analysis, though I'd appreciate a discussion of why you would make the estimate of a medium effect size. Do you have an example where you found such an effect size in this area of the literature? Further, in looking at your literature review, you note the importance of power on one null effect, but you actually report several null effects(e.g. refs 20 & 21). Given your finding and these findings, you can make a better estimate of effect size and you can calculate the power on the other null effects. Such an effort would certainly solidify the context of your findings and truly inform you if you had enough power.
Our RESPONSE: The reviewer is right that one main finding are the null results concerning emotion recognition. However, the second main finding, which refers to the eye-tracking data, indicates group differences in gaze behavior as a function of alexithymia and sex. We fully agree with the reviewer that more information is needed to understand why we assumed medium effect sizes in our group comparisons. We added more information on this point to our methods section (see p.4, l.192-202):
We expected group differences of medium effect size for eye-to-mask gaze ratio and emotion recognition performance. In her eye-tracking study, Fujiwara [2018] compared high and low alexithymic individuals and observed a medium-sized effect of group on eye preference (partial eta squared = 0.06). Parker et al. [1993] assessed recognition of facial expressions of basic emotions as a function of alexithymia: low alexithymic individuals showed a recognition score of 209.37 (with an SD of 54.79), whereas high alexithymic individuals had a recognition score of 180.18 (with an SD of 56.29). The latter results suggest a medium-sized effect of group (d = 0.53) regarding recognition of facial emotions in unmasked faces. Such an estimate of statistical power seems rather conservative for our experiment based on masked faces, since emotions are more difficult to identify in masked faces than in unmasked faces [Rinck et al. 2022].
- Why did you choose the emotions that you chose? You reviewed differences relevant to sadness and anger and the use of the eyes, but then you chose disgust and fearful. You accented the difference in cue location and cue utilization, but then didn't back up your choice with a rationale.
Our RESPONSE: We were interested in examining the perception of threat-related emotions as a function of alexithymia. In our revised introduction, we write (see p.2, l.74-87): “The systematic review of Grynberg et al. [38] concludes that alexithymic individuals’ impairments in recognizing emotions from facial expressions seem to be neither limited to specific emotional qualities nor a specific emotional valence. That means that alexithymia has been found to be linked to impairments in identifying negative (e.g., angry, disgusted, or fearful) expressions as well as to impairments in identifying positive (e.g., happy) expressions. However, there is evidence from more recent studies suggesting that alexithymia could be characterized by pronounced impairments in perceiving and processing fearful (Starita et al., 2018; Scarpazza 2014) and other threat-related facial expressions (Donges 2017). Disgust is a hostile emotion associated with aggression and conflict (Izard 1977) that expresses disapproval for the actions of other people (Haidt 2003), whereas facial fear is an indicator of indirect threat (danger in the environment) and the expresser’s loss of control (Wieser 2014). Against this background, it could be of particular interest to study the perception of facial fear and disgust along with that of positive and neutral expressions in alexithymic individuals.”
- I was surprised that you did not include an unmasked condition. Even just a small subset to balance out the other findings showing the same group differences reported in literature would have been very helpful. I don't think it would have taken too much time and given repeated measures, it would not have greatly hampered power. You could also explored the possibility that the full face might be more confusing since it offers too many sources of information instead of the masked condition which leaves only a smaller area of focus.
Our RESPONSE: We fully agree with the reviewer that it is a limitation of the present study that no data on emotion recognition in unmasked facial expressions are available. This is an important limitation of our study, which we acknowledge in our discussion section (see p.16, l.654-657).
- Alternatively, since the unmasked faces were of the standardized MPI faces you could have seen whether there was a difference from each group to the normed data of that data set. Some sort of baseline would also help your study assert what it to claim.
Our RESPONSE: We agree with the reviewer that the emotion recognition data reported in the validation study concerning the MPI faces of Ebner et al. (2010) can be used for a comparison with our recognition results (i.e., our hit rates). Moreover, the emotion recognition data of Carbon et al. (2022) appear also helpful to put our recognition data into a context, since Carbon et al. also administered MPI faces with face masks. We added the following paragraph to our revised discussion (see p.14, l.545-570):
“To evaluate the difficulty of our emotion recognition task we compared our overall hit rates (averaged across study groups) with hit rates of other investigations that also exam-ined emotion recognition based on the MPI faces database [22,86] in samples of young and healthy individuals using unmasked or masked facial expressions. Our hit rates for masked happy faces (0.87) and masked neutral faces (0.82) were lower than those reported by Ebner et al. [86] for unmasked young happy faces (0.98) and unmasked young neutral faces (0.93), averaged across young female and male raters. Our hit rates for masked fearful faces (0.91) and masked disgust faces (0.87) were higher than those reported by Ebner et al. [86] for unmasked young fearful faces (0.85) and unmasked young disgust faces (0.79), av-eraged across young female and male raters. Thus, for fearful and disgust expressions it can be noted that adding a face mask might have had a recognition-enhancing effect. At least for fearful faces, this effect could be due to the mask directing participants’ attention to a diagnostically highly relevant area, the eye region. When comparing our hit rates with those of Carbon et al. [22], who also investigated emotion recognition in faces with masks, similar hit rates were found for fearful faces (0.91 vs. 0.93). Our hit rates were higher for happy faces (0.87 vs. 0.75) and in particular for disgust faces (0.87 vs. 0.40) than those of Carbon et al. [22]. Finally, our hit rate for neutral faces was lower than that reported by Carbon et al. (0.82 vs. 0.95). All in all, it appears that in general our emotion recognition task could be less difficult than the task applied by Carbon et al. [22]. Even though both studies administered MPI faces our task required only the differentiation between four categories of facial expressions, whereas the task of Carbon et al. [22] required the differen-tiation between six categories of facial expressions. Moreover, it appears that the face masks applied in the study of Carbon et al. [22] covered more area on the upper nose and the lower eye region than the face masks in our study. These differences in face masking could have led to less recognition of disgust and happiness in the study of Carbon et al. [22].”
- Using Ekman's AU system with the prototypical faces, what AUs are still visible in the masked presentation mode? Is there a possibility that the eyes and brows are more diagnostic versus full set of available cues with the mask on across the emotion you selected? The forced choice task, which is fine for the purposes of this study, really requires identifying the relevant cues and the mask narrows the focus of cues. It's possible you are increasing the diagnosticity of the remaining cues - which is also why a baseline to unmasked faces would be helpful.
Our RESPONSE: It is unlikely that masking of the MPI faces (Ebner et al., 2010) made facial emotions in general better recognizable in in the masked compared to the unmasked face condition. Carbon (2020) and Carbon et al. (2022) also administered images of MPI faces in their studies, to which they applied face masks with a graphics editor. In both of these emotion recognition studies, MPI faces were shown with and without a mask. In both studies, recognition rates were not different for masked and unmasked fearful and masked and unmasked neutral facial expressions. In both studies, recognition rates were significantly lower for masked happy and for masked disgust faces compared to the unmasked face conditions (see Carbon (2020) and Carbon et al. (2022)).
- Any table that shows group differences should have subscript that identify which groups are different.
Our RESPONSE: We agree with the reviewer that for univariate ANOVAs it is appropriate to use superscripts to clarify which study groups manifest significant differences. We added superscripts to Table 1. In our opinion, superscripts can be misleading regarding ANOVA results with a repeated measure factor. In these cases, there must not exist significant between group differences for all levels of the repeated measure variable although the interaction effect is non-significant. Thus, against this background we decided not to use superscripts in the tables presenting repeated measures data.
- What are the correlations between DVs? A relationship between DVs is an opportunity to increase power and reduce the number of comparisons between IVs and DVs, the primary concern in Type I error. Can you consolidate? Regardless a table of correlations between IVs with each other and DVs with each other are very helpful.
Our RESPONSE: We agree with the reviewer that it could be interesting to present the correlations between (overall) recognition performance (hit rate, and RT for correct responses) and (overall) eyes-to-mask gaze ratio. These data can be useful to understand whether there exist associations between recognition performance and gaze behavior. However, in our opinion, it is not advantageous to present correlations between DVs for all expression conditions since this leads to multiple testing problems and the risk of false positive results. Moreover, the focus of the present study is not on the relations between emotion recognition and gaze behavior but on emotion recognition and gaze behavior in alexithymia.
Responding to the reviewer’s comment we added section 3.5. to our result section in which we describe the correlations between emotion recognition performance and eyes-to-mask gaze ratio in the whole sample, and in the alexithymic and non-alexithymic subsamples (see p.12, l.448-459). These correlations are shown in Table 6 of our revised manuscript. In none of the samples, correlations were found between emotion recognition (hit rate and response latency of correct responses) and eyes-to-mask gaze ratio. Thus, there was no evidence for a relation of hit rates or response times with an attentional preference of the eyes relative to the mask. In the whole sample and the non-alexithymic subsample significant negative correlations were observed between hit rate and response time. This means that high hit rates were as-sociated with fast responding in our emotion recognition task. In the alexithymic subsample, the correlation between hit rate and response latency of correct responses was also negative but non-significant (r = -0.21). This information can be found on p.12 (l.457-459) of our revised paper.
Moreover, we added a brief paragraph on this issue to our revised discussion section (see p.15, l.595-601), which reads: “In our study, visual attention to the eyes was not related to a better or faster recognition of facial emotions - neither in the whole sample nor in the alexithymic or non-alexithymic subsamples. It appears that participants did not benefit from longer fixation on the eye region (relative to the mask) concerning accuracy or speed of emotion identification in facial expressions. A possible explanation is that ceiling effects in the current study (low difficulty of our emotion recognition task) limited the ability to see a positive relationship of visual attention to the eyes and emotion recognition performance.”
- As I am looking at your literature review, your results, and some aspects of your discussion, it seems to me that there might be more than one path to alexithymia. One is through autism, an other might be through lack of verbal facility to label emotions. There may be others. I am particularly concerned about autism. Your exclusion criterion suggest no neurological disorder (which should mean no autism), but given your literature review should explicitly say including no autism. Further, did you screen with some quick screen tools that are now available. Autism is still under diagnosed and I would think you'd want to consider a score on a screening tool as a possible covariate - even if to quickly rule out.
Our RESPONSE: It appears rather unlikely that our study participants suffered from an autism spectrum disorder. We excluded participants with a diagnosed mental or neurological disorder from our study. Unfortunately, we did not assess autistic traits. This is a limitation of our investigation, which we acknowledge in the revised discussion section (see p.16, l.657-660). However, there is evidence that autistic and alexithymic traits are distinct phenomena. Recently, Cuve et al. (2022) using a factor‑analytic and network approach investigated whether alexithymia should be considered a product of autism or regarded as a separate condition. Their results support the claim that autism and alexithymia are distinct conditions. Against this background, we decided not to take up this issue in our discussion section.
- I want to highlight that as you have described it, Alexithymia is issue in the production and self-labeling of emotion and emotion recognition is the ability to see it in others. What is the mechanism by which these two processes are connected? That connection will speak to what you may or may not have found.
Our RESPONSE: In our view, there is a neurobiological connection between the perception of one’s own emotions and those of other persons. Brain regions involved in experiencing and feeling emotions are also thought to be crucial when processing facial expressions of others (Adolphs, 2002; Heberlein and Adolphs, 2007; Heberlein and Atkinson, 2009). Frontal and limbic areas play an important role in reenacting and feeling the according emotion, e.g., amygdala, striatum, frontal areas, and anterior insula (Adolphs, 2002; Heberlein and Adolphs, 2007; Trinkler et al., 2013). In addition, structures like the inferior frontal gyrus, insula, middle and superior temporal gyrus as well superior frontal and pre-motor areas are thought to be involved in the production and feeling of emotions, but also in recognizing these in others (Iriki, 2006; Van der Gaag et al., 2007). Given that the same regions are involved in recognizing one’s own emotions and those of others (Heberlein and Adolphs, 2007; Heberlein and Atkinson, 2009), it is well possible that individuals, who have difficulties in recognizing and describing own feelings also have problems to label facial expressions of others. However, as we did not assess brain functions in the present study, we prefer not to discuss our results in the context of the neurobiological shared substrate model in our manuscript.
.
- line 47 mentions "personality difference in emotion recognition." I have not heard it labeled as such. I think you are safer with individual difference.
Our RESPONSE: Following the suggestion of the reviewer, we revised the sentence in question, which reads now: “The role of individual differences in emotion recognition from occluded faces represents an important research question.” (see p.2, l.47).
REFERENCES
Adolphs, R. (2002). Neural systems for recognizing emotion. Curr. Opin. Neurobiol. 12(2),169–177.
Carbon, C.C. (2020). Wearing face masks strongly confuses counterparts in reading emotions. Front. Psychol. 11, 566886.
Carbon, C.C.; Held, M. J.; Schütz, A. (2022). Reading emotions in faces with and without masks is relatively independent of extended exposure and individual difference variables. Front. Psychol. 13, 856971.
Cuve HC, Murphy J, Hobson H, Ichijo E, Catmur C, Bird G. (2022). Are Autistic and Alexithymic Traits Distinct? A Factor-Analytic and Network Approach. J Autism Dev Disord. 52(5):2019-2034.
Donges US, Suslow T. (2017). Alexithymia and automatic processing of emotional stimuli: a systematic review. Rev Neurosci. 28(3):247-264.
Ebner, N.C.; Riediger, M.; Lindenberger, U. (2010). FACES - a database of facial expressions in young, middle-aged, and older women and men: development and validation. Behav. Res. Method. 42, 351-362.
Fujiwara, E. (2018). Looking at the eyes interferes with facial emotion recognition in alexithymia. J Abnorm Psychol 127, 571-577.
Grynberg, D.; Chang, B.; Corneille, O.; Maurage, P.; Vermeulen, N.; Berthoz, S.; Luminet, O. (2012). Alexithymia and the processing of emotional facial expressions (EFEs): Systematic review, unanswered questions and further perspectives. PLoS One 7, e42429.
Haidt J. (2003). The moral emotions. In: RJ Davidson, KR Scherer and HH Goldsmith, editors. Handbook of affective sciences. Oxford: Oxford University Press. pp.852–70.
Heberlein, A.S., Adolphs, R. (2007). Neurobiology of emotion recognition: current evidence for shared substrates. In: Harmon-Jones, E., Winkielman, P. (Eds.), Social Neuroscience,1st edition Guilford Press, New York, pp.31–55.
Heberlein, A. S., Atkinson, A. P. (2009). Neuroscientific evidence for simulation and shared substrates in emotion recognition: beyond faces. Emot. Rev.1(2),162–177.
Iriki, A. (2006).The neural origins and implications of imitation, mirror neurons and tool use.Curr.Opin.Neurobiol.16(6),660–667.
Izard CE. Human emotions. New York, NY: Plenum Press (1977).
Parker, J.D.; Taylor, G.J.; Bagby, R.M. (1993). Alexithymia and the recognition of facial expressions of emotion. Psychother. Psychosom. 59, 197-202.
Rinck, M.; Primbs, M.A.; Verpaalen, I.A.M.; Bijlstra, G. (2022). Face masks impair facial emotion recognition and induce specific emotion confusions. Cogn. Res. Princ. Implic. 7, 83.
Scarpazza C, di Pellegrino G, Làdavas E. (2014). Emotional modulation of touch in alexithymia. Emotion 14(3):602-10.
Starita F, Borhani K, Bertini C, Scarpazza C. (2018). Alexithymia Is Related to the Need for More Emotional Intensity to Identify Static Fearful Facial Expressions. Front Psychol. 11, 9:929.
Trinkler, I., Cleret de Langavant, L., Bachoud-Lévi, A.-C. (2013). Joint recognition-expression impairment of facial emotions in Huntington's disease despite intact understanding of feelings. Cortex 49(2), 549–558.
Van der Gaag, C., Minderaa, R. B., Keysers, C. (2007). Facial expressions: what the mirror neuron system can and cannot tell us. Soc. Neurosci. 2 (3–4), 179–222.
Wieser MJ, Keil A. (2014). Fearful faces heighten the cortical representation of contextual threat. Neuroimage 86:317–25.

Reviewer 2 Report
Comments and Suggestions for Authors
Author Response
Reviewer 2
In the article "Recognizing and Looking at Masked Emotional Faces in Alexithymia", the authors explored the recognition of emotions in faces with a face mask in high-alexithymic compared to nonalexithymic individuals and investigated the effect of alexithymia on attentional preference for the eyes when looking at emotional faces wearing face masks. The researchers conducted an eyetracking experiment in which they participants were displayed four types of faces with a superimposed mask. In the analyses, three metrics were taken into account i.e. hit rates, latencies of correct responses and fixation duration on eyes and the face mask. The results obtained by the researchers do not confirm the hypothesis that high-alexithymic individuals manifest impairments in the recognition of emotions in faces with face masks. The topics discussed in the article are interesting and useful as objects of research. The results, statistical analysis and discussion were properly and accurately described. Summing up, the paper is well written and therefore I recommend the submitted manuscript for publication after making a few minor corrections following the comments given below.
- At the end of the article, I suggest adding a conclusions section.
Our RESPONSE: Following the reviewer’s suggestion, we added a conclusions section to our revised manuscript (see p.17, l.674-685).
- I suggest writing the phrase "eye tracking" separately without a hyphen while "eye-tracking data" with a hyphen. This issue should be unified throughout the whole document.
Our RESPONSE: Responding to the reviewer’s comment we write “eye tracking” without a hyphen throughout the manuscript while writing "eye-tracking data" with a hyphen (see, e.g., p.1, l.11-12 and p.4, l.160).
- In line 90, I suggest using "technique" or "method" instead of the word "methodology".
Our RESPONSE: In our revised manuscript, we use the term “eye tracking technique” instead of eye tracking methodology (see p.3, l.107).
- In Figure 3, the Y-axis labels should be written with a decimal point, not a comma.
Our RESPONSE: As suggested by the reviewer, we use decimal points instead of commas for the y-axis labels in revised Figure 3.
- In the literature, double numbers in positions 14, 44, 48, 50 should be removed
Our RESPONSE: We removed the double numbers in our references.

Reviewer 3 Report
Comments and Suggestions for Authors
It was my pleasure to review the manuscript “ Recognizing and Looking at Masked Emotional Faces in Alexithymia” Results are very interesting and original. The present study had two main objectives. The first was to explore the recognition of emotions in faces with a face mask in high-alexithymic compared to non-alexithymic individuals. The second objective was to investigate the effect of alexithymia on attentional preference for the eyes when looking at emotional faces wearing face masks.
Abstract
The abstract is well-written and concise.
Introduction
Well written. It would be helpful to discuss similarities and differences between autistic and alexithymic participants in emotion recognition from occluded faces. The link between mood, verbal intelligence, visual motor processing, and emotional recognition should be elaborated. Line 144-149
Material and methods
Are well written.
Results
No statistical data is provided in tables 1-4, 6-7. It will be helpful to observe significant results and statistical data in the tables.
The discussion is well written. The sex difference in alexithymia participants is well discussed, but the other factors are hardly mentioned in the discussion.
Conclusion Not addressed.
Limitations of the study addressed.
Author Response
Reviewer 3
It was my pleasure to review the manuscript “ Recognizing and Looking at Masked Emotional Faces in Alexithymia” Results are very interesting and original. The present study had two main objectives. The first was to explore the recognition of emotions in faces with a face mask in high-alexithymic compared to non-alexithymic individuals. The second objective was to investigate the effect of alexithymia on attentional preference for the eyes when looking at emotional faces wearing face masks.
Abstract
The abstract is well-written and concise.
Introduction
Well written. It would be helpful to discuss similarities and differences between autistic and alexithymic participants in emotion recognition from occluded faces. The link between mood, verbal intelligence, visual motor processing, and emotional recognition should be elaborated. Line 144-149
Our RESPONSE: In our introduction of our revised manuscript, we point out that in the study of Pazhoohi et al. [21] extreme groups were compared, and we specify their results: high autistic trait individuals manifested worse facial emotion recognition than low trait autistic individuals when faces wearing masks were presented but the extent of this impairment was similar when unmasked faces were shown (see p.2, l.50-53). In the paragraph, where we describe the null results of two previous studies that investigated the correlations between alexithymia and emotion identification in faces wearing face masks (Verroca et al. and Maiorana et al.) we underline the importance of extreme group comparisons (see p.2, l.97-99): “Previous research on the effect of autistic traits shows that an extreme group approach can be important to detect recognition impairments in the recognition of emotions from masked faces [21].” Moreover, we note in our introduction that “Autistic and alexithymic traits could modulate distinct aspects of face perception: autistic traits were found to be associated with structural encoding of faces, whereas alexithymic traits were related to emotion decoding processes [Desai 2019].” (see p.2., l.54-57)
Moreover, following the reviewer’s suggestion, at the end of our revised introduction we present more information on the relations of negative affectivity and verbal intelligence with emotion recognition and we present a rationale why we administered the TMT-B in our study (see p.4, l.162-169).
Material and methods
Are well written.
Results
No statistical data is provided in tables 1-4, 6-7. It will be helpful to observe significant results and statistical data in the tables.
Our RESPONSE: We decided to revise Table 1 and to add superscripts to it. In this way, readers can realize which means differ significantly from each other. However, in our opinion, superscripts can be misleading regarding ANOVA results with a repeated measure factor. In these cases, there must not exist significant between group differences for all levels of the repeated measure variable although the interaction effect is non-significant. Thus, against this background we decided not to use superscripts in the tables presenting repeated measures data. Furthermore, we feel that since our tables with results from four study groups are rather filled up with information it would not be very helpful to bring statistical data into the tables (and double the statistical information within the manuscript). It should also be noted that several tables show non-significant data concerning the factor group (Tables 2, 3, and 7).
The discussion is well written. The sex difference in alexithymia participants is well discussed, but the other factors are hardly mentioned in the discussion.
Our RESPONSE: In our discussion section we focus on the effect of alexithymia and biological sex on emotion recognition and gaze behavior. This requires a lot of space in the discussion (already about 2500 words). The variables verbal intelligence, visuomotor processing speed, depressive symptoms, state and trait anxiety were not associated with gaze behavior, and affectivity was not related to hit rate or response latency. To save space, we decided not to discuss these null results. However, in our revised discussion, we compared our overall hit rates (averaged across study groups) with hit rates of other investigations that also examined emotion recognition based on the MPI faces database [Ebner et al. 2010, Carbon et al. 2022] to evaluate the difficulty of our emotion recognition task (see p.14, l.545-570). Moreover, in our revised discussion we discuss the fact that in our study visual attention to the eyes was not related to a better or faster recognition of facial emotions - neither in the whole sample nor in the alexithymic or non-alexithymic subsamples (see p.15, l.595-601). Finally, we added study limitations to our discussion section (see p.16, l.654-660).
Conclusion Not addressed.
Our RESPONSE: We added a brief conclusions section to our revised manuscript (see p.17, l.674-685).
Limitations of the study addressed.
References
Carbon, C.C.; Held, M. J.; Schütz, A. (2022). Reading emotions in faces with and without masks is relatively independent of extended exposure and individual difference variables. Front. Psychol. 13, 856971.
Desai, A; Foss-Feig, J.H.; Naples, A.J.; Coffman, M.; Trevisan, D.A.; McPartland, J.C. Autistic and alexithymic traits modulate distinct aspects of face perception. Brain Cogn. 2019, 137, 103616.
Ebner, N.C.; Riediger, M.; Lindenberger, U. (2010). FACES - a database of facial expressions in young, middle-aged, and older women and men: development and validation. Behav. Res. Method. 42, 351-362.
Maiorana, N.; Dini, M.; Poletti, B.; Tagini, S.; Rita Reitano, M.; Pravettoni, G.; Priori, A.; Ferrucci, R. The effect of surgical masks on the featural and configural processing of emotions. Int. J. Environ. Res. Public Health 2022, 19, 2420.
Pazhoohi, F.; Forby, L.; Kingstone, A. Facial masks affect emotion recognition in the general population and individuals with autistic traits. PLoS One 2021, 16, e0257740.
Verroca, A.; de Rienzo, C.M.; Gambarota, F.; Sessa, P. Mapping the perception-space of facial expressions in the era of face masks. Front. Psychol. 2022, 13, 956832.

Round 2
Reviewer 3 Report
Comments and Suggestions for Authors
I was pleased to review the manuscript “ Recognizing and Looking at Masked Emotional Faces in Alexithymia”. My recommendation would be to accept the manuscript in its current form for publication